## Classics

art and science; chromatin organisation; chromosome painting; cytogenetics; interphase nuclei.

**Corresponding author:**
Alexandre Berr;
Email: alexandre.berr@ibmp-cnrs.unistra.fr

# The art of painting chromosome loops

Alexandre Berr and Marie-Edith Chabouté

Institut de Biologie Moléculaire des Plantes (IBMP), CNRS, Université de Strasbourg, Strasbourg, France

## Abstract

How to get a metre of DNA into a tiny space while preserving its functional characteristics? This question seems easy to pose, but the answer is far from being trivial. Facing this riddle, salvation came from technical improvements in microscopy and *in situ* hybridisation techniques applied to cytogenetics. Here, we would like to look into the past at one of these pure cytogenetics articles that makes a breakthrough in addressing this question in plant science. Our choice fell on the work published two decades ago by Fransz et al. (2002). Besides the elegant manner in which DNA probes were organised to bring into light the out-looping arrangement of interphase chromosomes in *Arabidopsis thaliana* nuclei, this article perfectly illustrates that painting is not reserved to the fine art. As for whether emotional expression prioritised by artists can sometimes hide behind scientific empirical evidence, there is only a small step to make to the general case.

Besides their typical X-shaped portrayal observed during mitosis, chromosomes undergo an elaborate, variable and non-random 3D folding pattern inside interphase nuclei. This packaging and organisation primarily depend on the bendability of the DNA, an ability directly related to its stiffness properties. Besides this, different binding proteins and interacting partners will fold, wrap and loop genomic DNA. A classic example is the tight coiling of DNA around the histone octamer (two each of histones H2A, H2B, H3 and H4), forming a nucleoprotein complex named nucleosome, the fundamental unit of the chromatin fibre. Interestingly, this chromatin fibre is compartmentalised inside the interphase nucleus without being delineated by membrane-based structures. Indeed, the regulatory machinery of all DNA-related processes (i.e., transcription, replication, and DNA repair) are architecturally organised in subtle nuclear domains with specific dedicated functions (Shao et al., 2018). Thus, direct and reciprocal relationships between nuclear structures and functions were predicted to exist with cells following some similar and main key rules (Cremer & Cremer, 2010). Exploring these relationships has enabled the identification of a nearly endless number of multi-scale prominent domains within the nuclei of plants and animals, including, among others, chromocenters, chromosome territories, nucleolus, heterochromatin, nuclear matrix, nuclear lamina, Polycomb bodies, RNA polymerase II factories, Cajal bodies, different types of speckles and the Barr body observed in somatic cells of certain female animals (Finn & Misteli, 2019).

In 1885, the cytologist Carl Rabl first proposed a territorial arrangement of interphase chromosomes, far from the early belief of their lawless intermingling as spaghetti pasta in a bowl. Based on microscopic observations of dividing cells in two amphibians (*Salamandra maculate* and *Proteus anguinus*), Rabl hypothesised that the polarised orientation of chromosomes established during anaphase remains during interphase, with centromeres and telomeres being located at opposite poles of the nucleus (Rabl, 1885; Figure 1a). Twenty years later, the concept of chromosome territory with an individual chromosome occupying a distinct and spatially limited volume in the nucleus was settled based on cytologic studies in the plant *Galtonia candicans* by Eduard Strasburger (Strasburger, 1905; Figure 1b) and, few years later, in the parasitic worm *Ascaris megalocephala univalens* by Theodor Boveri (Boveri, 1909; Figure 1c). It was only in the 1980s that, following the introduction of *in situ* hybridisation techniques into cytogenetics, the concept of a chromosome territorial organisation was corroborated (Cremer & Cremer, 2010). In this context, the method described by Bauman et al. in 1980 (Bauman et al., 1980) has marked a major milestone in molecular cytogenetics and has been at the genesis of fluorescence *in situ* hybridisation (FISH). FISH is a powerful method based on the use of fluorescently labelled oligonucleotide probes for the spatial and also quantitative detection of nucleic acids on

cytogenetic preparations with applications in both genomics and transcriptomics. From there, progressive improvements in probe design and labelling using various fluorophores, as well as in signal amplification using secondary reporters, have rapidly made it possible to apply FISH for whole chromosome visualisation approaches. The 'useful art' of chromosome painting was born (Ried et al., 1998). Using numerous specific parts or entire chromosomes-based probes, chromosome painting offered for the first time the possibility to visualise, simultaneously and with high specificity and sensitivity, several individual chromosomes during metaphase or interphase (Huber et al., 2018). Consequently, in the early 1990s, chromosomes, which was a term derived from the Greek 'chroma' for colour and 'soma' for body (Waldeyer, 1888), truly became multi-coloured when Cremer et al. (1988), Lichter et al. (1988) and Pinkel et al. (1988) independently performed the first human chromosome painting experiments.

Meanwhile, efforts to initiate the establishment of FISH in plants overall yielded unsatisfactory results. One reason for this was assumed to likely rely on the large amount of repetitive DNA sequences spread across plant genomes and retrieved in chromosome-derived probes (Fuchs et al., 1996). By dint of perseverance (as always in science...), the solution came from the ever most well-known plant model *Arabidopsis thaliana*, whose small genome size (1C = ~135 Mb), small chromosome number (n = 5) and relatively low amount of repetitive DNA sequences (about 20%, while it reaches more than 80% in species like maize) appeared as assets for FISH. Another key point was the progressive release during the 1990s of physical maps covering the entire *A. thaliana* genome and made of thousands of ordered yeast or bacterial artificial chromosome (YAC or BAC) contigs containing large DNA inserts (i.e., average size ranged from 100 to 1 Mb; Choi et al., 1995; Creusot et al., 1995; Mozo et al., 1999). Finally, at the turn of the new millennium, Fransz et al. (2000) succeeded for the first time in painting specific regions of *A. thaliana* chromosome 4 short arm using five YAC clones (Fransz et al., 2000). In the wake of this advance, the first specific painting of an entire euploid plant chromosome was achieved shortly thereafter by the group of Prof. Ingo Schubert on *A. thaliana*, using labelled pools of BAC contigs individually selected for the absence of dispersed repeats as probes for FISH on meiotic chromosome spreads and also on interphase nuclei (Lysak et al., 2001). Together, beyond obtaining high-resolution integrated cytogenetic maps, these works have perfectly illustrated the potential of FISH for comparative mapping of euchromatin and/or heterochromatic segments in plants, but also for the study of chromosome organisation during interphase.

From there remained only a step that the next paper by Fransz and colleagues (Fransz et al., 2002) shortly spanned over. Owing to the recent FISH technical improvements, the choice of *A. thaliana* as an experimental model just seemed obvious. Yet, this choice was also dictated by the need to explore the most fundamental mechanisms behind the organisation of DNA inside interphase nuclei. The use of a species with a fairly simple chromosomal set like *A. thaliana* (i.e., compact genome packed into only five (sub)metacentric chromosomes with conspicuous heterochromatin clustered mainly at centromeric/pericentromeric regions and organising regions abbreviated as NOR at terminal nucleolus) has therefore appeared as a preliminary and necessary step. Another major strength of this work was that *A. thaliana* chromosomes can, in theory, 'easily' be distinguished by FISH using distinct and specific heterochromatic probes targeting the centromeric 180-bp repeat pAL, the 5S rDNA or the 45S rDNA (NOR). By experimentally applying this last hint and using distinct

labelling, Fransz and colleagues revealed the compartmentalisation of the terminal NOR on chromosome 2 and 4 short arms together with their respective centromere at the nuclear periphery of interphase nuclei where they form masses of dense heterochromatin called chromocenters. Hence, it was concluded that these short arms arrange themselves in megabase-long loops starting on and ending at a unique heterochromatic chromocenter. As an additional piece to the *A. thaliana* chromosome organisation puzzle, the number of distinct chromocenters was found on average to be around 8–9 per interphase nuclei with a maximum of 10, thus suggesting their almost systematic mutual repulsion. More functionally and using immunolabelling, Fransz and colleagues also noticed that chromocenters contained heavily methylated DNA, while being deprived in acetylated histones, known to be synonymous with active transcription. Even if it was not really a surprise from the DNA-dense aspect of these nuclear foci after DAPI staining, this result further supports their transcriptional inactivity. Then, applying FISH on the telomeric sequence, they observed that telomers preferentially localise around the nucleolus. As a wink to past cytogeneticist glories, *A. thaliana* was then officially introduced into the growing club of species with a non-Rabl chromosome configuration. But then, what lurks behind this non-Rabl pattern? The strategy to address this question was both very simple and incredibly clever. Fransz and colleagues designed probes from contiguous BACs along the short arm of chromosome 4 (i.e., one of the two *A. thaliana* chromosomes carrying a NOR at the end of their short arm). The key was that BAC probes were pseudo-coloured alternatively in green and red along the short arm following a precise sequence (Figure 1d). After hybridisation on interphase nuclei, all that remained was simply to read the colour sequence starting from the DAPI-dense chromocenter while assuming that any change would in fact mean that additional smaller euchromatic loops do exist within the megabase-long original loop. This ingenious trick together with the occurrence of colour inversions at a high frequency laid the foundation for the chromocenter-loop model. Finally, using up to a few hundred BACs, Fransz and colleagues further deduced from FISH patterns that each interphase chromosome territory in *A. thaliana* is organised as a well-defined heterochromatic chromocenter located at the nuclear periphery from which transcriptionally active euchromatin loops of variable length (0.2-2 Mb) emanate, ultimately forming a complex and dynamic network of chromatin interactions. Taken as a whole, the 2002 paper by Fransz et al. has provided compelling evidence about the 3D organisation and behaviour of interphase chromosomes in plant nuclei at an astounding resolution. With around 20 citations per year, this seminal and avant-gardist cytological work has since inspired many other studies on diverse plants and has been supported by high-throughput methods, super-resolution imaging microscopy and computational modelling (for review see Jiang, 2019; Randall et al., 2022).

From its origin in 1842 when Karl Wilhelm von Nageli first observed chromosomes in pollen (Vines, 1880) to now, cytogenetics has been regarded, and rightfully so, to be as much a basic science as an art form. Indeed, how can one not see in Rabl, Strasburger and Boveri's drawings the hand of the artist (Figure 1a, b and c). Similarly, while the concept of chromosome painting expanded in the work of Fransz et al. (2002) may seem a purely scientific one, it is not difficult to feel the artistic intent behind it. Exactly like how an artist would use paint to create an artwork, chromosome painting involves colouring chromosomes in different shades. From the word 'painting', but also from the aesthetics

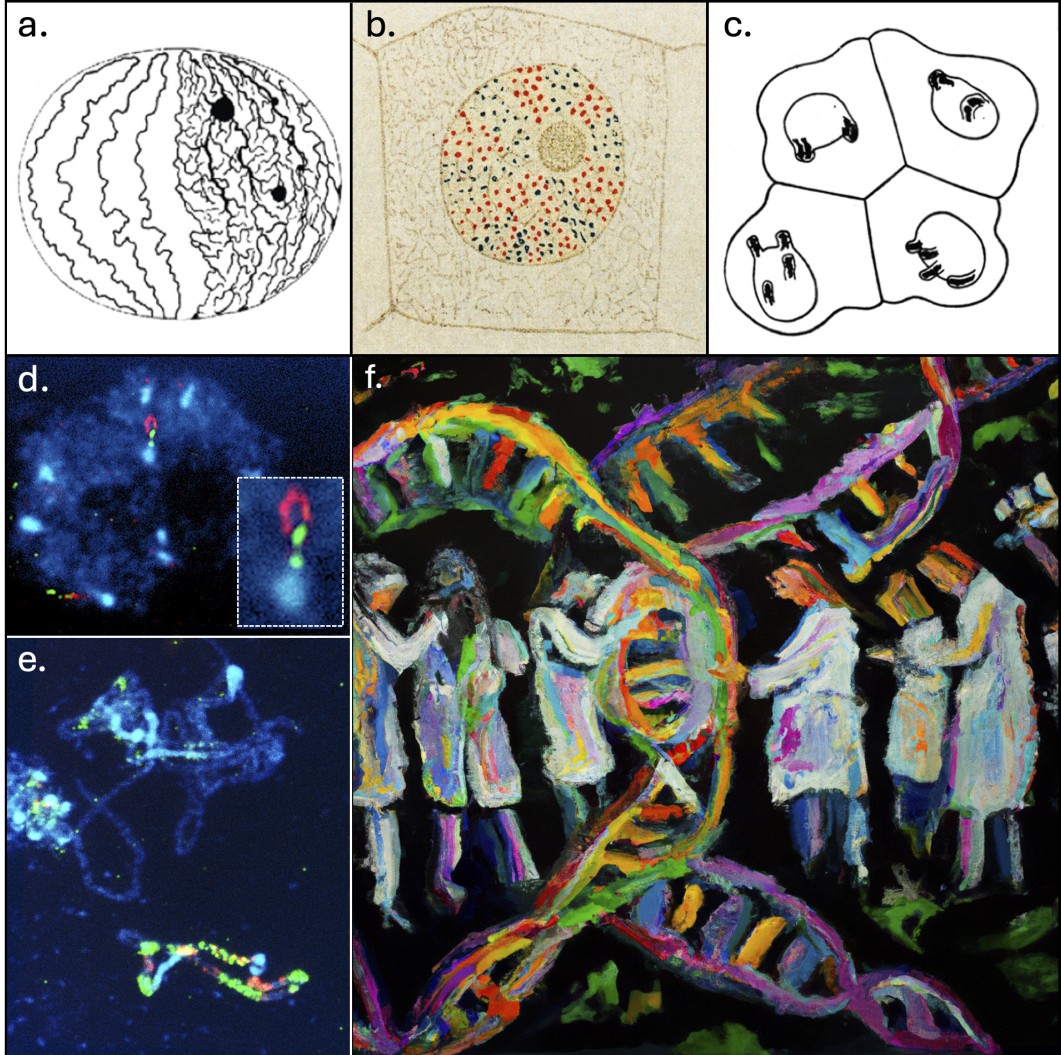

**Figure 1. The art of cytogenetics for chromosome arrangement.** (a) The Carl Rabl view of the territorial arrangement of interphase chromosomes with telomers on one side (bottom) and centromeres on the other (top). (b) Artistic coloured drawing by Eduard Strasburger of a nucleus with chromosome territories made of higher-order foci in blue and red dots. (c) Theodor Boveri's view of the distinct or symmetrical arrangement of nuclear protrusion between two pairs of distinct daughter cells (two upper and two lower ones). (d) Adjacent BACs labelled in red and green and hybridised at the end of *Arabidopsis Thaliana* chromosome 4 top arm form a small loop structure (reprinted from Fransz et al., 2002. Copyright 2002, National Academy of Sciences, U.S.A.). (e) Chromosome painting of pachytene *A. thaliana* chromosomes hybridised with a pool of BAC probes covering the entire chromosome 4 bottom arm and differently labelled in red and green (Paul Fransz, personal communication). (f) Artistic view of what chromosome painting is as a contemporary confluence between science and art. This illustration was generated by the OpenAI system DALL-E2 (https://openai.com/dall-e-2) on 31 March 2023, using the following description: 'Several scientists in white lab coats painting in several colors a chromosome on a black canvas with flashy colors. Use an impressionist style with a black background'.

of the resulting figures, scientists prove to us (intentionally or not) how this technique can have artistic merit (Figure 1d, e and f). Digging down behind this odd connection, science and art may *stricto sensu* be considered as non-intermingling domains (Kemp, 2005). Science often implies objectivity, while art better matches with subjectivity. Thus, drawn on their respective approaches, scientists are usually used to rigorously and precisely value empirical evidence, while artists prioritise boundless personal expression and emotion. In keeping with the intrinsic distinctive characters between science and art, Pablo Picasso said in an interview with the magazine *The Arts* (1923), '*We all know art is not truth. Art is a lie, but it is a lie by which we know the truth.*' Just like a mirror to Picasso's words, Lorenz Konrad, one of the founders of modern ethology, wrote that '*Scientific truth is universal, because it is only discovered by the human brain and not made by it, as*

*art is*' (Lorenz & Fritsch, 1977). Nevertheless, even if science and art do not record their own truths about the world in the same way, they are both united in a shared curiosity for the unknown. This shared curiosity needs imagination and passion to evolve, also suggesting creative excellence as the combination of natural talent and idiosyncratic character. Further emphasising the unification of science and art, Luigi Pirandello, the 1934 Nobel Prize winner for literature, elegantly wrote that '*every work of science is both science and art, and each work of art is both art and science*' (Pirandello, 1908). Based on observations and interpretations, scientists and artists create a new world, the first by changing the way we interpret it and the second by changing the way we see it. For centuries, the confluence between science and art has constantly given rise to new ways of thinking about our world and interacting with it, always pushing boundaries of what is possible and the strength

of our existence over and over. Many stunning or more discrete examples of such art–science confluence exist (Zhu & Goyal, 2019), but we will only mention a few of them here. During the 13th century, the mathematician Leonardo of Pisa described in his book *Liber Abaci* the Fibonacci sequence (1, 1, 2, 3, 5, 8, 11, …) as a quest of beauty in mathematics and of mathematics in the beauty around us (Rozin, 2020). This sequence is found all throughout nature, from the spiralling shapes in artichoke, cauliflower and sunflower, to the spiral patterns on seashells. In art, the Golden ratio (1:1.618033988) derives from the Fibonacci sequence and was used throughout centuries to produce attractiveness, equilibrium and harmony, as visible in *The Creation of Adam* by Michelangelo to *The Great Wave* by Katsushika Hokusai, or in the abstract paintings by Piet Mondrian to the *Vitruvian Man* by Leonardo da Vinci. It is precisely da Vinci, during the Renaissance, who excelled in crossing disciplines and who based his artwork related to the human body on preliminary anatomical studies. Moving forward four centuries, how could one fail to mention Santiago Ramón *y* Cajal, the father of modern neuroscience, as being, in addition to a fanatically precise observer, an exceptional artist. To account for this, there is nothing like leafing through the aesthetically magnificent art book *Beautiful Brain*, gathering together some of Cajal's most remarkable illustrations (Newman et al., 2017). The same is true for the 19–20th-century biologist Ernst Haeckel who found beauty even in the most unlikely sea creatures, as testified in his legendary portfolio (Haeckel, 1998), or for the 20th-century physicist Richard Feynman, who, besides his contribution to the Manhattan Project and the quantum mechanics, was also an accomplished and talented drawer and painter (Feynman et al., 1997).

In recent years, the frontier between art and science has remained tenuous. Nevertheless, we obviously lack hindsight to judge, in the light of the past, about their actual confluence as the Yin and Yang of culture (Goldberg, 1997). As scientists, we continuously keep pushing the boundaries of our understanding. But in this era of hyper-specialisation, where information, technology and knowledge are moving too fast, we should never forget that the world surrounding us holds plenty of aesthetic charms. We should never forget also that we have the responsibility to communicate our work directly to the public, and for that, we need creative and transformative holistic approaches. Thereupon, who better than artists would be used to it? On the one side, artists have an insatiable thirst for inspiration, and on the other side, scientists excel at generating ideas. While scientific communication should not be limited simply to making aesthetic covers for scientific magazines by means of digital tools, singular initiatives do exist to substantially promote the connection between art and science, making artists the new interpreters of scientific discoveries (Williams, 2017). One of these is the "Art of Science" exhibition, which is held annually at Princeton University. The purpose of this exhibition is to enhance the accessibility and engagement of scientific research to a broader audience by showcasing artworks that draw inspiration from it. Given that a person who is acquainted with both art and science should find it natural to combine the two, why not encourage more art/science initiatives in academia or simply offer new courses to train experts in both disciplines? People like David Goodsell, whose watercolour paintings are tightly linked to his own studies, could then inspire new vocations (Goodsell, 2021). Hence, despite obstacles that might hinder the confluence of science and art, there exists considerable potential for the two disciplines to continue to cooperate and mutually stimulate one another, fostering a deeper understanding of our world. Notwithstanding the inquiries and sometimes even the apprehensions it evokes, artificial intelligence (AI) could act as an efficient catalyst to speed up the pace of the modern synthesis between art and science ("Collaborative creativity in AI", 2022; Figure 1f). In closing, consider this modest essay as a simple reminder that when distinct fields collide, some potential necessarily emerges.

## Acknowledgments

We would like to warmly thank Professors Marion and Thomas Cremer, who kindly shared with us pictures of original drawings from Carl Rabl (Figure 1a), Eduard Strasburger (Figure 1b) and Theodor Boveri (Figure 1c), as well as Professor Paul Fransz for providing FISH pictures from his original article (Figure 1d and e). We also thank Etienne Herzog and Gilles Dupouy for useful and heated discussions on the topic.

**Financial support.** This work was supported by the Centre National de la Recherche Scientifique (CNRS), by ANR-18-CE20-0011 project REWIRE and ANR-20-CE13-0025 project MechaNUC.

**Competing interest.** The authors declare no competing interests exist.

**Authorship contribution.** A.B. and M.E.C. wrote the article.

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
