## [Reviewer Report]

Dear Alexandre Berr,

Thank you for your patience in bearing with lengthy editorial runtime. The delay was due to the scarcity of reviewers who could adequately evaluate your manuscript. Please accept our apology in handling your manuscript not within an expected time frame. I could readily see your manuscript is highly original and good fit for Quantitative Plant Biology (QPB) that aspires to embrace and promote interdisciplinary articles. With comments from one reviewer and my own assessment as an editor/reviewer, we would like to proceed to publish your article at QPB. We highly value your proposition and insights on the intersectional venue of arts and science.

Before moving on, I would like to check one issue with you, regarding the image usage. I had already consulted with the journal for publishing AI-generated image as well as using old images. We would really appreciate if you can add some notions on any copy right issues regarding image sharing as well as some more details on AI-image generation. Hopefully, the following details could help you add up to clarify the issue. Please also address all the comments from the reviewer in the revised manuscript.

1) QPB’s policy on AI usage

Any use of an AI tool to generate images within the manuscript should be accompanied by a full description of the process used and declared clearly in the image caption(s). Descriptions of AI processes used should include at minimum the version of the tool/algorithm used, where it can be accessed, any proprietary information relevant to the use of the tool/algorithm, any modifications of the tool made by the researchers (such as the addition of data to a tool’s public corpus), and the date(s) it was used for the purpose(s) described. Any relevant competing interests or potential bias arising as a consequence of the tool/algorithm’s use should be transparently declared and may be discussed in the article

2) For the old images in Figure 1

Unless this is taken from a recent book, this should be public domain now. Otherwise, the authors can ask for the copyright (rightslink). It’s usually free for an academic journal.

3) I had independently confirmed with Paul Fransz for the use of his original FISH image. No need to act on this image, as you already clearly stated on this.

4) Typo to fix, L51: remove space between fiber and “.”.